# Public–Private Partnerships Model Applied to Hospitals—A Critical Review

**DOI:** 10.3390/healthcare11121723

**Published:** 2023-06-12

**Authors:** Nuno J. P. Rodrigues

**Affiliations:** 1Research on Economics, Management and Information Technologies, REMIT, Portucalense University, Rua Dr. António Bernardino Almeida, 541-619, 4200-072 Porto, Portugal; nuno.rodrigues@islagaia.pt; 2School of Management, ISLA—Polytechnic Institute of Management and Technology, Rua Diogo Macedo n.º 192, 4400-107 Vila Nova de Gaia, Portugal

**Keywords:** PPP, healthcare quality, care pathways

## Abstract

In this paper, a conceptual framework for investigating the PPP model as it relates to hospitals is proposed. When the PPP model is applied to healthcare (hospitals), it is possible to discover the path to success by developing a critical assessment and deriving a clear model. It is concluded that most PPP model implementations in hospitals around the world have produced favorable outcomes, both in terms of the performance of healthcare units and in terms of cost-effectiveness. Additionally, a path-to-success model that applies to hospitals is offered, taking into account six PPP model dimensions: (i) Environment; (ii) Potentiate Benefits; (iii) Constant Measure; (iv) Evaluation; (v) Management; and (vi) Enhance Strengths. The PPP model only applies case by case and under specific requirements that should be met cumulatively to provide additional value to healthcare’s quality of service. The right conditions are created, the right benefits are amplified, public concerns are frequently assessed, private contributions are carefully considered, and all pressing challenges are managed by enhancing both public and private strengths. Leading decision- and action-making processes in corporate, governmental, and social sectors is the goal of managing PPP models.

## 1. Introduction

A public–private partnership (PPP) is an agreement between one or more public and private entities, usually of a long-term nature, which reflects mutual responsibilities in the promotion of common interests [1]. Ref. [2] further provides a description, defining PPP as a type of structured cooperation between public and private partners in the planning/construction and/or operation of infrastructure, in which they share or redistribute risks, costs, benefits, resources, and responsibilities. Additionally, in the United States of America, the National Council for Public–Private Partnership defines PPPs as contractual arrangements between a public sector agency and a private for-profit company, in which resources and risks are shared for the provision of a public service or development of public infrastructure [3,4,5].

The present research is justified by the fact that there is an open debate about PPPs’ estimated gains and the benefits obtained through their use [6,7,8], particularly in the health area (e.g., hospitals). Additionally, the PPP management model is very controversial among academics [6,9,10]. This translates, in many situations, into a dissonance between purely economic–financial analyses and the perception of quality and satisfaction with the service, especially when making decisions (e.g., award or renewal) regarding a specific PPP project. In addition to the open debate and controversy present in the scientific community, due to the dearth of studies examining the effects of PPPs on the effectiveness and quality of the services delivered, there is a significant knowledge gap according to [7]. The author contends that there is little information available to decision makers about the conditions under which the PPP model is likely to yield favorable outcomes. Despite the fact that the PPP model is very popular among academics, the literature that is currently available is still fragmented and primarily focuses on the management cost component. However, especially in the health sector, a holistic analysis of hospital performance is required and not only a simple cost analysis [11].

Given the growing number of alternatives developed for healthcare delivery, the PPP model being one of them, the search for appropriate evaluations is a key factor for the development and empirical benchmarking of solutions in healthcare. As a contribution to the literature, we attempted to propose a model that will allow practitioners to evaluate the extent to which the PPP model might be applicable to a specific healthcare system (hospitals). The research is organized with regard to the following research questions:Is it possible to identify the clear path to success while applying the PPP model to hospitals?It is possible to identify an ideal scenario where the PPP model would undoubtedly be of added value to the quality of healthcare?

## 2. Methodology

Literature reviews are often lengthy pieces that can be used to describe and discuss the progress or "state of the art" of a specific subject from a theoretical or contextual standpoint. An explicit and transparent approach based on the principles of methodologically pluralistic research, or a mixed method, should be used in any rigorous literature review [12,13,14]. The need to carry out the synthesis of present knowledge stems from the vast number and fast pace of scientific publications [15] related to PPP model application in healthcare services, namely hospitals. Additionally, ref. [16] argues that conducting a literature review in organizational and management studies is particularly challenging given the fragmented nature of the area.

Taking this idea into consideration, a critical review was conducted focusing on the main scope “PPP applied to Hospitals” leading to a set of key words applicable to this research [17]. The present research included the keywords ”PPP”, ”Health Care”, and ”Hospitals” in some form of the words individually or in combination, adopting the steps proposed by [18], combined with the snowballing technique [19] and principles of problematization research [20]. This process made it possible to find the key texts for additional reading. Given that the search phrases (keywords) govern the parameters and nature of the literature search, they were carefully chosen to permit the selection of all pertinent articles while simultaneously excluding irrelevant ones [15].

The critical review method is used to develop existing or produce new hypotheses or models. Since systematic reviews frequently provide quantitative answers, critical reviews differ from them. This approach is used to assess prior research and competing theories to offer a foundation for conceptual development [21]. It describes an author’s hypothesis or conceptual model based on the literature in the aimed scope [22]. According to the same authors, one of the objectives is to show that the reviewer has a solid comprehension of the literature to the point where they may deduce conceptual models or hypotheses about the research issue, going beyond the level of in-depth description of the existing literature. Taking the aforementioned ideas into consideration, the critical review methodology fits the research objectives.

## 3. PPP Model Theoretical Framework

PPPs are marketed as win–win agreements and are distinguished by their lengthy lifespan and significant private sector funding [23,24]. The literature contends that the most desirable compromise between price and the quality of the infrastructure and accompanying services, rather than the lowest price, should be used to compare PPP projects to traditional public procurement [25]. The PPP model allows organizations to attract financial and technical capabilities in areas where the public sector is not as strong [10]. On the other hand, ref. [5] acknowledge that PPPs are not a miracle solution in the context of infrastructure services (e.g., hospitals) and that their value needs to be considered and assessed on a case-by-case basis.

In order to evaluate the model on a case-by-case basis, it is crucial to identify the advantages and disadvantages of the model. Ref. [4] identified the essential advantages which include: (i) the sense of enhanced partnership between public and private sectors; (ii) better risk management; (iii) the implementation of clearer government policies; (iv) the identification of critical success factors; (v) improved contract management; and (vi) more appropriate financial analyses. Particularly, for the public partner, the main advantages are improved performance, cost-effectiveness, better service delivery, and appropriate allocation of risks and responsibilities. The private partner, on the other hand, strives for better investment potential, reasonable profit, and more opportunities to expand its business interests [5,26]. The dissonance in the literature regarding the application of the PPP model is mirrored by the advantages and disadvantages of the model presented in Table 1.

Besides advantages and disadvantages, it is also important to understand that one dimension that must be assessed is related to the model’s public accountability and its application to government and private institutions. Thus, ref. [35] proposed an analytical framework that aimed to assess to what extent the PPP model provides (or will provide) goods and services in accordance with public sector objectives regarding its effectiveness, efficiency, and equity. The assessment is measured in six dimensions: (1) risks, (2) costs and benefits, (3) social and political impacts, (4) skills, (5) collaboration, and (6) performance measurement. For [35], managers’ strategic thinking regarding net gains for the public partner obtained by offering the model and careful consideration of the six-dimensional framework should result in a professional management structure that ensures their constant monitoring throughout the life cycle of the partnership, thus leading to a management structure that reflects a culture of public accountability.

Management activities play a critical role in determining the success or failure of a PPP project, referring to a dimension of jointly determined norms and rules designed to regulate and control individual and group behaviors [36]. PPP model management is about guiding decision-making and action processes in private, public, and social sectors. Typically, PPP projects have a theoretical framework and a set of key questions that guide their management. From the systematization of the literature, it was possible to discover that the most pressing issues related to the PPP model management are cooperation, trust, communication, capacity, risk allocation and sharing, competition and, finally, transparency [36].

### Model Applied to the Health Sector

Health sector policymakers are increasingly engaging in PPP projects, in which private companies are contracted by governments to finance and deliver new hospitals and related services [7]. To define the term PPP applied to the health sector, authors of previous publications [37,38,39] used the health system and the respective provision of services as a basis. Ref. [37] addresses the PPP model in terms of private sector contributions to the public sector, quantifying them at three levels:i.Greater efficiency, either by reducing the cost of services or by increasing the quality and effectiveness of services;ii.Expanded access to health services for the entire population, using more cost-effective means of providing services and using the specialized resources of the private sector;iii.Production of additional resources and revenues for the public sector, through the rental of facilities/equipment or specialized knowledge to the private sector at reasonable costs.

For [37], the PPP model represents a repositioning of health systems by governments, uniting the logic of effectiveness and efficiency, focusing on a three-dimensional objective: (i) to allow public entities to access private capital and better management; (ii) to allow the public sector to benefit from the experience of the private sector and for the state to access technologies in a less costly way and, above all, to redistribute risks, the security of service provision, and the extension of health infrastructures and services; and (iii) to use, in an intelligent way, the flexibility and exactitude of the private sector to counterbalance the slowness of regulations and procedures in the public sector, thus allowing a better adjustment of the state’s means to its objectives [38].

Finally, ref. [39] argues that, in a context where there is an accumulation of financing needs and an increased supply of health services and various budgetary constraints, the PPP model aims to improve the effectiveness and efficiency of public services and strengthen financial, operational, and infrastructural viability. Contrary to previous assertions, ref. [39] argues that private funding of clinical services, instead of allowing the public sector to free up time and resources, contributes to its depreciation through the enactment of mixed funding regimes, inferring more difficult management and accounting processes [40].

Ref. [39] establishes a benchmark for fresh lines of inquiry by illuminating the feasibility of rigorous assessment and comparison of quality across PPP contracting methods. Ref. [11] also makes the case that, despite the PPP model’s high level of interest in academia, the material that has been published so far is still fragmented and primarily focuses on the topic of management expenses.

Ref. [41] recognized the key industries where nations will need to boost and enhance their supply, with the health sector representing one of the opportunities for private sector involvement. The current literature has already produced some systematic reviews that summarize the knowledge in the scope of PPP model applied to health (e.g., [7,27,42]). By analyzing the reviews available, it is possible to argue that PPP combines the strengths of private partners (e.g., innovation, technical knowledge and skills, management efficiency, and entrepreneurship) with the strengths of public partners (e.g., social responsibility, social justice, public accountability, and local knowledge), thus creating an enabling environment for the provision of high-quality health infrastructure and services [42]. Additionally, through the application of PPP, both partners can achieve benefits such as job creation and educational development in the field of innovation and competition and the development of health infrastructure [42]. However, [27] argued that there is no understanding of the main factors and features that lead to the success of the PPP model, identifying an increasing number of problems related to the implementation of this type of model. Ref. [27] argues that the government must act as a regulator in industries such as healthcare where accountability is crucial and general welfare is at stake. Thus, more active government participation is required to mitigate issues that emerge with the implementation of PPP applied to hospital construction projects, namely issues related to cost, quality, flexibility, and complexity. Refs [7,27] argued that PPPs in the health sector should not only fill the gaps in the provision of services to the population and dominate the development of infrastructure allocated to healthcare, but should also offer several other potential benefits, such as:Enable existing infrastructure to be used more efficiently;Enable health programs, run by governments, to be accelerated;Promote motivation of both health professionals and patients;Some parts of the population need to be considered as markets; this can be established through PPPs.

However, projects lack the rigorous risk analysis, optimal risk allocation, and focus on whole-project cost and long-term performance management required to achieve these benefits.

The influence on the public interest, according to [27], is one of the key issues with PPPs in the healthcare industry. According to the author, real-world examples of the model’s use in the health sector show a decline in the quality of service delivery. The underfunding caused by the expenses of PPPs, which push service provision cuts to reduce deficits and harms the public interest, is the cause of this decline in quality. The authors suggest that in order to safeguard public interest, the guiding principles of PPP efforts should focus on the idea of equity in health and be focused on the benefits to society rather than the partners’ mutual advantage.

Empirical evidence suggests that PPPs are promising for decision makers who wish to obtain greater certainty about outcomes, such as cost, quality, and volume of services, than can be obtained through alternative public procurement mechanisms [7]. On the other hand, where government capacity is limited, PPPs allocated to services provided in hospitals are unlikely to produce good outcomes, in terms of cost and quality certainty. Thus, there is theoretical consensus on the benefits that the PPP model can bring when applied to healthcare. However, the factors leading to the success of partnerships are not clear and the scientific evidence raises some doubts regarding the increase in quality measured by the PPP model’s application. Notwithstanding the analysis carried out on systematic literature reviews, it is important to detail other studies present in the literature.

Ref. [43] analyzed a PPP model involving the construction of a hospital and the continuous provision of its clinical and non-clinical services within a public healthcare delivery system. The author argued that where there is little contestability and performance evaluation problems, the service should be provided within a public management hierarchy; conversely, where performance measurement and evaluation is straightforward and contracting is highly contestable, the service, or good, should be procured using the private sector. Despite the justification offered, the advantages of the PPP model over the conventional contracting approach are still hotly debated. It is feasible to name a number of critical concerns that continue to cast doubt on the benefits of the PPP model, including cost, quality, flexibility, and complexity. Following this idea, exploring the dimension ”Cost versus Quality”, ref. [43] presented positive conclusions when comparing the PPP model to public procurement. The author concluded that the construction of new hospitals is more likely to be executed on time and within budget when a PPP model is applied; however, these gains often appear to be at the expense of quality. Ref. [43] also concluded that it is impossible to say whether the PPP model is flawed or whether lower quality results from poor construction execution. One plausible interpretation is that the added complexity of the PPP model makes all but the simplest projects too diffuse.

Ref. [6] concluded that regulatory stability, employment rate, hospital capacity, construction duration, and concession period have a significant relationship with total infrastructure investment. In a general way, the authors concluded that a higher level of value for money (*VfM*) can be achieved in PPP hospital projects when they are developed within a good economic and political environment to stimulate competition and, in turn, decrease public expenditure. Additionally, it is concluded that small hospitals, which require less construction effort, when granted for long periods, are more likely to achieve better operating results, bearing a lower level of construction risk. Along the same line, ref. [44] examined assumptions about the PPP activities and its role in improving public procurement of surgery and primary healthcare. The authors argued that the success of PPP projects depends on the ability of private participants to (i) assess how PPP assumptions add value to their activities; (ii) assess the effectiveness of those assumptions applied to the intended activities, and (iii) prepare to align their business principles with PPP government objectives. Ref. [44] concluded that giving public partners more discretion than private partners over critical decisions can help ensure that the assumptions in PPP activities culminate in outcomes that match the anticipated health benefits. They also concluded that PPP features alone may not adequately explain why and how they may not produce benefits when applied to healthcare services. Indeed, private partners bring important new skills and expertise, but how they interpret the assumptions of their activities facilitates their performance, making them much more useful in achieving the anticipated health benefits.

Ref. [29] set out to study how ”Big Data” models can improve the quality and timeliness of information in PPP healthcare infrastructure investments, making them more sustainable and increasing their overall efficiency. Despite the benefits already noted, it is feasible to pinpoint the following primary problems with the application of the PPP model in this study: (a) complexity, in terms of designing the transaction and writing the necessary documentation; (b) higher borrowing costs compared to conventional financing; and (c) lengthy negotiation of financing and operating agreements. According to the study, collaboration between public and private partners enhances value co-creation, boosts *VfM*, and lowers risk when big data is networked and interoperable databases are used. Additionally, the authors argued that big data is available in massive terms from different sources and in real time and is likely to have a remarkable impact on the planning and management of PPPs in the health sector, with continuous feedback and tuning that reduces risk and improves value for money and resilience to external shocks.

Academic understanding of the PPP model’s application to healthcare continues to be a hot topic of debate. There is a significant lack of actual data to back up the claims made about PPPs, despite the fact that theoretical and conceptual evaluations show significant improvements in terms of the quality of services delivered and cost savings.

## 4. The Performance of the PPP Model

Regarding how to measure PPP model performance, the literature (e.g., [25]) outlines three categories that should be measured when implementing and operationalizing PPP projects:*Costs*—refers to operational costs, capital costs, or total project life cycle costs;*Quality*—refers to the infrastructure quality, e.g., in terms of functional use or quality perceived by users;*Vfm*—refers to providing public goods in sufficient quantity and quality, improving efficiency and value for money, fulfilling the social objectives it set out to achieve.

Regarding the ”Costs” dimension, ref. [25] found that nine projects, developed under a PPP regime, resulted in a higher cost compared to conventional public procurement. Three projects resulted in lower costs and, finally, another three projects showed dubious results. These findings guided the author to question the rhetoric of the proponents of the PPP model, which identify the model as a mechanism to achieve greater cost efficiency and lower costs during the life cycle of infrastructure construction projects. The ”Quality” dimension also revealed mixed results, with some PPP projects presenting superior performance and others inferior performance, when compared to traditional public procurement. However, 18 of the 21 projects examined by [25] do not assess service quality, which made it difficult to obtain any solid conclusions. Regarding the ”Value for Money” dimension, the evaluation can be performed using three different approaches: (1) ex ante evaluation of the PPP model vs. business case; (2) evaluation of the PPP vs. public sector comparator (PSC); and (3) evaluation of the PPP against (real) benchmarks of similar projects. Despite the consensus in the literature as to the applicability of these three approaches of *VfM* evaluation, the same does not applies to the results obtained by each of them. Considering the advantages and disadvantages of each *VfM* evaluation approach, the one based on the evaluation of the PPP model in relation to project reference values is understood as preferential, compared to the first two approaches [25]. Comparing the conventional procurement alternative with PPP resulted in mixed conclusions, with two projects demonstrating a higher *VfM* when the PPP model was applied and two studies indicating the opposite. Thus, ref. [25] argued that the empirical evidence does not support the theoretical expectation concerning cost savings resulting from task integration and private sector expertise, as claimed by PPP proponents.

On the other hand, ref. [45] analyzed the results of four PPP hospitals in Portugal, concluding that the PPP model, applied to the health sector, appears to be advantageous, not only regarding economic and financial results, but also regarding the quality of service provision. The same authors also concluded that the patients of the PPP hospitals were more satisfied than the users of the public hospitals. Ref. [46] also concluded that, between 2012 and 2017, PPP hospitals in Portugal provided health services with performance levels at least as good as those provided by public hospitals. These findings reinforce the idea, referenced in the literature, that private partner management practices can deliver services at a lower cost [47] and in an economically efficient manner ([35,48]).

Ref. [26] stated that social and technical variables are crucial to guaranteeing the viability of PPP projects and are, thus, the determining factors for successful performance and, subsequently, for a project’s success. The author considered success factors and stakeholders’ concerns. It should be emphasized that gathering performance indicators is an important stage in carrying out likelihood of success research for PPP projects, and the crucial success elements can act as the deciding factors for PPP feasibility evaluation [26]. These evaluation factors can be classified into five categories:Technical Factors—The PPP model is not considered as an attractive or viable option if the requirements and technologies change continuously during the expected duration of the project;Financial and Economic Factors—for the partnership to be attractive to investors, a PPP project must be self-sustaining, financially viable, and profitable, which in turn is highly dependent on the economic environment, government policy, and the inflexibility of competition;Social Factors—social acceptance is indispensable in today’s society; thus, the government should never choose a PPP model to improve facilities or service delivery without meeting the demands and expectations of the population. [26] argued that citizens are more cautious about the quality of service and costs incurred when facilities or services are delivered through PPP models;Political and legal factors—lack of political support is considered as a potential obstacle to PPP projects; thus, a PPP model may be rejected if it is considered as politically sensitive. Any changes in the political environment or deviations in the legal framework increase uncertainties and increase the risk of failure of a PPP project;Other Factors—human resource issues and government management actions.

It should be mentioned that articles assessing management tools and their application may be found dating back to 1989 (e.g., [49]). Ref. [49] concentrated on how decision makers both inside and outside of government view management tools, and more specifically, on the standards used to assess the instruments’ viability for addressing policy issues. According to the author, public decision makers may base their choices on which tools to employ on custom, instinct, ideology, or a simple acquaintance with them. Ref. [49] concluded that a specific instrument that was poorly created and used in a certain situation would be useful as a starting point for further research into other proxies. However, by itself, such a conclusion does not offer anything about the prior factors that influenced the initial instrument selection.

Considering the constraints revealed above regarding the choice of management instruments by governments, it should be noted that among the incentives that organizations (public or private) face are the need for change in the creation and modification of products or services, the adaptation to new technologies or the improvement of existing ones, the change in strategy or organizational culture, and the introduction of new management models and processes [50]. Supporting this idea, ref. [51] argued that management plays a critical role in the public, private, and third sectors of the economy, with each sector requiring definitions of strategies and goals to be attained, the development of people, the measurement of performance, and the marketing of a specific organization’s products and/or services. New global demands and an increasing need for infrastructure building and recovery (the "infrastructure gap") have compelled governments to turn to private partners in order to reduce the funds included in their respective state budgets while still being able to meet growing investment needs [52].

## 5. Empirical Research on PPP Applied to Healthcare

We examined a wide range of works in the literature to evaluate the state of the art and the empirical findings surrounding the implementation of the PPP model in the health sector. In this way, it was possible to comprehend the model’s achievements and failings when applied to hospitals as well as the directions for future research advised by works of the literature.

In order to determine whether the PPP model is a viable alternative that enables the performance of public hospitals to be improved, ref. [40] conducted an empirical study on how PPPs are perceived in public hospitals in Romania. No matter which PPP typology is used, according to the authors, unless the interests of users are protected and the private sector is encouraged to find a balance in managing the partnership with efficiency and dynamism, PPPs will not improve the performance of public sector institutions. Regardless of the number of beds, the number of patients, the indicators of episode complexity, the typical waiting time for operation, or the amount of ward cost, ref. [40] came to the conclusion that PPPs in Romania increased the performance of public hospitals.

Regarding the communication issue, ref. [53] glanced into PPPs in Canada and concluded that there was a lack of communication with the populace when a new hospital was opened under a PPP regime. Overall, the authors came to the conclusion that the PPP project under study had a number of major issues, but none of them directly related to the private sector’s involvement through the adoption of the PPP model. Ref. [54] developed a health infrastructure index and a healthcare delivery index that were applied to several Indian states. The use of the PPP model, according to the authors, appears to have significant potential to break this impasse. The authors argued that deficiencies in terms of comprehensiveness, access, and affordability, together with rising healthcare expenses, have aggravated the social crisis in India. The same authors also concluded that, unless it is made available to the people who need it most, health infrastructure (such as hospitals) as such does not necessarily guarantee the accomplishment of governmental objectives. The authors claimed that the PPP model, when used in the health sector, is an effective way out because it is unlikely that the problem can be solved with public resources alone.

In the context of comparing the two models (PPP hospitals vs. public hospitals), ref. [11] asserted that the coexistence of the two models in the same region requires a thorough investigation to apply rational standards for determining which model is most appropriate. As a result, the authors conducted a comparison between public hospitals and the PPP model (Alzira) in order to pinpoint any potential strengths and flaws in both models and lay the groundwork for logical judgments regarding the applicability of each. The outcomes demonstrated that the PPP model had some advantages over public hospitals, exhibiting superior performance in some of the examined categories. Additionally, it was discovered that PPP hospitals enhance some quality indicators, such as the decline in wait times on the waiting list. The authors did discover several places where this kind of collaboration had less clarity, though. Thus, the results were not conclusive enough to clearly acclaim the PPP model, due to the discovery of strengths and weaknesses in both public and PPP models.

The PPP model used for hospitals in Spain recently came to an end, imposing an ex post examination of the model. Ref. [55] assessed the PPP model’s performance between 2003 and 2015, collaborating the findings with all public peers in the NHS (National Health Service), and compared the life cycle performance of the PPP model with other similar service providers. Ref. [55] concluded that the PPP model performed statistically worse than the benchmark in most measures in 2015 when compared to public peers. However, it had one of the best performances in the NHS in terms of adjusted mortality after percutaneous coronary intervention. The authors came to the conclusion that, despite certain notable technological advancements, the PPP model did not consistently outperform public providers. Ref. [56] conducted a study in which the PPP model was applied to hospitals in Portugal and evaluated after the conclusion of the PPP project, providing valuable results and a brief glimpse into how PPP hospitals and state hospitals may be contrasted. The PPP model, when implemented in hospitals, was advantageous according to the author both in terms of the quality of services offered and the economic and financial outcomes. The author also concluded that the patients of the PPP hospitals are more satisfied than the users of the public hospitals. Despite the findings, the author emphasized that regardless of the PPP model’s additional value, the position and choice regarding its application always appear to depend on the political force in power. Finally, ref. [56] contended that the Portuguese government consistently chose to switch from a PPP model to a public model despite the value for money produced by the PPP model for hospitals under analysis, the cheaper construction costs, and the improved performance in comparison with comparable public hospitals.

Additionally, in the context of international evaluations of the PPP model, ref. [57] investigated the challenges associated with facility management service delivery under PPPs for health services in Malaysia. The study produced a list of fourteen issues that must be resolved before using PPP models for hospitals. The most major obstacle is described as developing a specialized model appropriate for PPP risk management, followed by the difficulty in securing agreements and the complexity of the project. The authors argued that a PPP agreement for the management of a hospital can ensure added value and have a positive impact on service quality and the corporate image of the health sector. The authors outlined the difficulties that partners in this kind of relationship confront, including those related to risk, cost, complexity, design, responsibility, and contract formalization. Thus, ref. [57] recommend that stakeholders, who already have or are considering engaging in PPP projects for hospitals, should initially address and mitigate the identified challenges to avoid any inefficiency in service delivery. Ref. [58] advocated that the finance phase is crucial for differentiating between various infrastructure investment strategies. The author concluded that if an investment has a high level of technological uncertainty and the hospital asset has a high level of specificity, public hospitals with strong bargaining power appear to select a PPP-oriented model. This suggests that the uncertainty of technological change plays a significant role in the occurrence of an outcome and decision that favors a PPP-oriented model of asset investment. Given the above findings, there are two robust combinations of factors that favor the PPP model. The first combination is the low specificity of human resources combined with considerable technological change uncertainty and weak negotiating leverage. High technological uncertainty, high asset specificity, private ownership, and weak negotiating power make up the second strong combination [58]. Ref. [59] showed the influence of implementing the PPP model in hospitals, particularly in Russia, arguing that the benefits of the PPP model are clear, with innovative technologies helping to decrease the number of deaths, reduce the number of people with disabilities, and increase life expectancy and working capacity of the population. Additionally, it was concluded that the results showed that the PPP model has some economic success. In the same scope, ref. [60] analyzed the policy objectives of the PPP model applied in hospitals, focusing on the experience of the ”East Azerbaijan” province of Iran. The study’s findings suggested that the PPP policy’s primary goals are to increase access to healthcare in underserved regions of the nation and decrease the need for extraordinary payments. According to the results of the study, the main reason for the design and implementation of the PPP model in the country’s healthcare service was the inefficiency of the public sector in effectively providing health services to the entire population.

Regarding new opportunities and barriers, ref. [9] analyzed and assessed the barriers and opportunities for improving the state of public hospitals in Poland through the PPP model. The authors made the case that there are several opportunities for cooperation between public and private sectors, such as when private and public partners work together in a hospital. The PPP model, however, has substantial obstacles, including short-term contracts with public funds and significant underfunding of healthcare. Ref. [9] concluded that the main factors that can affect the success of PPP in the Polish market are (a) changes to the contract with the payer of healthcare services; (b) stability of economic and legal conditions; (c) the inappropriate allocation of risk; (d) sufficient experience of both parties; and (e) use of a reputable and competent private partner with sufficient initial capital. The authors argued that the primary driver behind the creation of PPP models is financial motive. Direct or indirect financial methods may be used by private businesses, but for public hospitals, reconstruction and creative management techniques generally should assist in reducing costs and enhancing hospital services. Thus, if properly formulated and managed, PPPs can provide several benefits to the public health sector, and two of the most important seem to be a reduction in the financial burden on the public sector for infrastructure development and the sharing of risks between partners. By examining different experiences of different PPP models in various countries, ref. [9] argued that legal, cultural, and social influences are of great importance for the success of partnerships. Additionally, the authors argued that the implementation of PPPs in the public hospital sector, despite some controversy, has positive implications and has become widely accepted.

Finally, it is important to analyze the study carried out by [61] in China. The authors note that by the end of 2018, China’s local governments owed 18.4 trillion yuan, and there was intense pressure to increase social health spending. By using the PPP model, the study sought to investigate the variables influencing the intention and behavior of private sector involvement in the Chinese NHS. Ref. [61] conclude that the Build–Operate–Transfer (BOT) model may promote the enthusiasm of private capital participation in the Chinese NHS; this is because several private sector companies are mostly directed towards construction activity.

## 6. Discussion and Conclusions

There is no doubt about the diversity of opinions on the PPP model’s theoretical discourse. While the theoretical benefits of the model and the likelihood of favorable outcomes are presented on the one hand, some skepticism about the outcomes of the PPP model’s application is starting to surface on the other. There are conflicting results in the current study’s focus on hospitals, which indicates that varied political, economic, and social situations could lead to both successes and failures in this field. It was possible to conclude that most PPP model implementations in hospitals around the world have produced favorable outcomes, both in terms of the performance of healthcare units and in terms of cost-effectiveness. It is true that there are a number of elements that should be taken into account when choosing this contracting model, including public interest, the difficulty of the project, and public outreach (particularly when discussing PPP models). Further, the financial benefits of PPPs show that average cost savings can reach 20% [9]. Although using the PPP model in healthcare is inevitable, it is important to understand that one of the main issues with the model in this industry is how it affects public interest since private partners may not always have public interest as their primary concern. In other words, PPPs can be understood as either public policy in action or as solely commercial partnerships with an economic interest. This implies that a continuum of ethical and political behavior, from trust in integrity to unethical and calculated activity, can exist in a partnership. There is no doubt about the diversity of opinions on the PPP model’s theoretical discourse. If the PPP model theoretically predicts several benefits (mutual) and the possibility of successful financial outcomes, the results attained through the deployment of the PPP model are mixed. The present study’s focus is on the health sector, and there are conflicting findings in this field as well, leaving room for the possibility of both successes and failures stemming from various political, economic, and social situations. Therefore, within this framework, it is essential to try to draw a clear path describing the steps to be taken to guarantee the success of PPP projects used in hospitals.

Regarding the defined research questions, we proposed a model (Figure 1) wherein a path to success is presented, taking into consideration six dimensions associated to the PPP model applicable to hospitals: Environment; Potentiate Benefits; Constant Measure; Evaluation; Management; and Enhance Strengths. The dimensions were defined taking into consideration all the empirical evidence analyzed throughout this research.

Nevertheless, regarding the road to success, how can we identify an ideal scenario where the PPP model would undoubtedly be of added value to the quality of healthcare? The ideal scenario needs to be set with clear roles performed by both partners. Figure 1 highlights the importance the roles, asserting that the public partner should have a regulatory and supervisory role. In this sense, the conclusions presented by [62] are relevant. The authors argued that the government failed to effectively supervise and monitor the PPP model, preventing it from acquiring new management skills. They also claimed that the government failed to change the level of service expected of public management hospitals to that of private partnership hospitals. So, if the defined role is not correctly undertaken by the public sector, all the concerns identified in the proposed model (Figure 1) will be amplified. *VfM* is one of the main concerns for the public partner, being the optimal combination of the life cycle costs of the project and the quality of the service aimed at meeting user requirements (optimal combination of whole life cycle costs, risks, time to completion, and ability to meet public requirements) [63]. Thus, where the PPP model delivers *VfM*, there will be a positive difference between the PSC and the value of the PPP tender. The PSC can be defined as the best project that can be undertaken and financed directly by the state through public procurement to perform, with all the specified requirements, the provision of the service and achieve the same objectives as the process developed in a PPP regime [27,41].

Another concern identified in the proposed model is public interest in healthcare provision. The guiding principles of PPP initiatives should be founded on the potential benefits to society rather than the mutual benefit to partners and should be centered on the concept of equity in health to preserve public interest. The advantages of the PPP model over the public contracting paradigm, however, are still hotly debated. Cost, quality, flexibility, and complexity are just a few of the pressing concerns that persist. However, it was assessed that a higher level of *VfM* can be achieved in PPP hospital projects when they are developed within a good economic and political environment to stimulate competition and, in turn, decrease public expenditure.

To date, empirical evidence has not supported the theoretical expectation of cost savings from task integration and private sector expertise, which is why the PPP model is applied to hospitals as a means of achieving greater efficiency and lower costs during the life cycle of construction and the delivery of health services [25,42]. Theoretically, a private investor assuming ownership of a public hospital will have a negative impact on patients’ contentment and the standard of care they receive [64,65]. From the current research, it is possible to argue this idea, namely by taking into consideration the potentiated benefits achieved in Portugal [45], Spain [55], Iran [60] and Russia [59]. It should be noted that there are differences in the quantity and quality of the services offered in the health sector as well, necessitating the assessment of these differences and making the process of determining the *VfM* more difficult because the quality of the services does not directly translate into an economic measure [66]. These conclusions leave open the possibility of there being cases of success and failure, resulting from different political, economic, and social environments. So, it is possible to conclude that political and social environments (satisfaction and quality of service) represent the most critical dimensions in terms of their weight in a possible final result. Therefore, the PPP model should only be appropriate in cases where a number of conditions are cumulatively met in order to undeniably improve the quality of services provided by the healthcare system. I.e., the right environment is set, benefits are correctly potentiated, public concerns are regularly measured, private contributions are deeply evaluated, and all pressing issues are managed through an enhancement of both public and private strengths.

## Figures and Tables

**Figure 1 healthcare-11-01723-f001:**
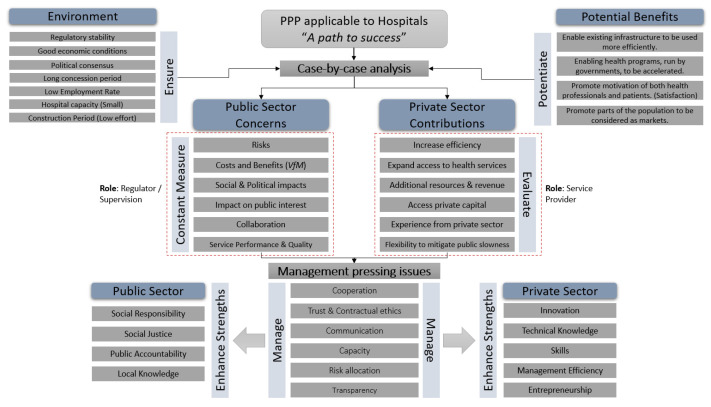
Proposed model.

**Table 1 healthcare-11-01723-t001:** Summary of advantages and disadvantages of the PPP model.

Advantages of the PPP Model [4,6,7,27,28]	Disadvantages and Problems of the PPP Model [2,29,30,31,32,33,34]
Contracts are accompanied by a robust regulatory acquis, which implies a strong regulatory and monitoring component.	Problems inherent in the whole negotiation process, including greater contractual rigidity, may condition changes to the project. Overly aggressive private sector use and poorly designed contracts can lead to financial imbalances and ultimately greater scrutiny for profits.
It makes it possible to attract financial and technical capacity, in areas where the public sector is not as strong.	Higher costs of borrowing compared to conventional finance.
Allows the state to access new sources of funding to address budgetary constraints.	It may be more expensive, as the private sector tends to express the risk explicitly by including a risk premium in its price and the private partner may try to optimize the life cycle cost of the project, which may result in an increase in the initial investment.
The sense of enhanced partnership between the public and the private sector.	Conflicts between public and private interests, particularly regarding the pursuit of *VfM*, with distinct objectives for both partners, can end up in fuzzy contracts and generate ”noise” in the management of the investment.
Enables better risk management.	Doubts surrounding the effective transfer of risk between parties.
Promotes the implementation of clearer government policies.	
Allows the identification of critical success factors.	Negotiation of financing and operating agreements tends to be time-consuming.
Encourages improved contract management.	Difficulties of supervision of the contracts established by the public partner.
Promotes more appropriate financial analysis.	Doubts about the quality of the services provided.
Allows existing infrastructure to be used more efficiently.	
Enables health programs, run by governments, to be accelerated.	
Promotes motivation, both of health professionals and users.	
Some parts of the population need to be considered as markets; this can be established through the PPP model.	

Source: Own elaboration.

## Data Availability

No new data were created or analyzed in this study. Data sharing is not applicable to this article.

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
