# Peer review of "Public–Private Partnerships Model Applied to Hospitals—A Critical Review"

_healthcare, 2023, doi:10.3390/healthcare11121723_

Round 1
Reviewer 1 Report
The subject of PPPs remains current. The presented article adds value to the knowledge already produced.
The proposed model is coherent with the review and adjusted to the questions defined.
The sources used are relevant, and their analysis seems carefully directed.
English should be improved.
Following up on the request, here are the details of the issues raised:
1. Briefly summarize the content of the manuscript;
The article explores the development of a conceptual model for studying and applying the PPP model to Hospital units. Interestingly, the author(s) attempted to assess the subject topic based on a critical review applying a snowballing technique and principles of problematization research. The overall findings of this article indicate a path to success for the PPP model, taking into consideration six dimensions associated with the PPP model applicable to hospital units. Additionally, the author(s) attempted to identify an ideal scenario where the PPP model would be a undoubtedly solution to increase the quality of service in the healthcare area.
2. Illustrate what are, in your opinion, its strengths and weaknesses (this is an essential step, as the editor will consider the reasoning behind your recommendation and needs to understand it properly);
Strong contribution: The author(s) clearly point out which parts (based on evidence) are different from previous research, namely when identifying the current gap in the literature. As the author(s) move forward with their critical revision, it becomes clear the entire scope related to the PPP model applicable to Hospital units, explicitly the “Management of the PPP model" and "Health sector around the world”. Despite the dispersion of positions regarding the theoretical rhetoric of the PPP model identified by the author(s), the proposed review gave rise to a clear path to success whereas the PPP model is applied to healthcare, namely to hospital units. Another substantive strength is about clear methods applied. The author(s)' approach, regarding the analysis of the PPP model applicable to hospitals, elaborating a critical review in order to develop a new conceptual model, contributes to further clarification regarding the optimal conditions to maximize the PPP model results. Finally, the evaluation of the existing literature and competing ideas has a strong relationship that allowed a starting point for conceptual development. I think that this manuscript has an interesting approach and has in-depth theoretical and practical implications.
Weak points. The author(s) summarizes the advantages and disadvantages of the PPP model taking into consideration a vast literature, but I think that a more descriptive analysis might be necessary (which I admit is complex and difficult to execute given the necessary limitations in terms of content) . Also, the author(s) argue the PPP theory taking into consideration recent empirical research results on four different countries. In my opinion, the results can pose a challenge to the PPP theoretical framework but the author(s) could have explored the idea further despite the good scientific approach.
3. Provide a point-by-point list of your major recommendations as to what must be improved.
English should be reviewed at the level:
- Excessive use of the passive voice;
- validation of UK and USA English;
- Reduction of the length of some sentences.
Please review the use of passive voice.
Please verify the spelling of words like healthcare or health care (with space) (11, 212, ...).
Check commas, dots and the use of articles and prepositions.
Author Response
We thank the reviewer for the opportunity to improve our paper.
English should be reviewed at the level:
- Excessive use of the passive voice.
[Nuno Rodrigues]: Many thanks for the remark, we completely agree. Passive voice errors were corrected. Track changes are available to follow the corrections made.
- validation of UK and USA English.
[Nuno Rodrigues]: Many thanks for the remark, we completely agree. All the article was corrected to be in English UK.
- Reduction of the length of some sentences.
[Nuno Rodrigues]: Many thanks for the remark, we completely agree. Sentences length corrected. Track changes are available to follow the corrections made.
Reviewer 2 Report
This scholarly article delves into the application of public-private partnerships (PPPs) in the healthcare sector, particularly within hospitals, through a critical review methodology. Despite the abundant research in this field conducted and documented in the extant body of literature, this reviewer contends that the topics explored in this paper remain pertinent and applicable to developing and developed economies. Nevertheless, several pressing concerns require the author's attention to ensure the acceptability of this paper, as elaborated upon below.
a. Firstly, this paper requires meticulous proofreading to address grammatical or typographical issues. Numerous errors are perceptible within the text, and they can be distracting to the reader. It is recommended that the author seeks the assistance of a professional proofreader to address this concern.
b. The second noteworthy concern pertains to the structure of the paper, which renders the content and context challenging to grasp, at least from this reviewer's perspective. Despite the extensive use of references, they appear to be used without a clear connection or coherence. As a result, the critical review by the author is not apparent within the paper. To address this issue, the author may consider reorganizing the paper's structure and establishing a clear and logical flow of ideas supported by relevant references.
c. The paper's primary research questions were twofold: i) can a clear and definitive path to success be identified when implementing the PPP model in healthcare, particularly in hospitals? ii) is it possible to pinpoint an ideal scenario where the PPP model would unquestionably enhance healthcare quality? However, this reviewer deems that the paper has not fully addressed both inquiries. It is recommended that the author reorient the approach and conduct a comprehensive critical review focusing explicitly on these two questions, enabling them to achieve the research objectives.
d. Still related to point c, the first research inquiry is partially answered in the paper's conclusion section, albeit without a clear explanation of its reasoning (i.e., jumping to conclusions). Concerning the second research question, the statements expressed in lines 553-557 are deemed insufficient in their justification and do not adequately respond to the question. The author is advised to revisit the conclusion section and reinforce the arguments with robust reasoning in the discussion section (not in the conclusion section), providing a comprehensive answer to both research questions.
e. This reviewer further argues that certain conclusions presented in the paper are inconsistent with the cited references and the author's statements. On lines 492-494, the author acknowledged the presence of contradictory findings in the area of hospitals that "leaves open the possibility of there being cases of success and failure, resulting from different political, economic, and social contexts." However, on lines 498-500, the author concluded that "most have shown positive results" regarding healthcare unit performance and cost efficiency. This conclusion appears to be in conflict with the earlier statement. Therefore, it is recommended that the author revisit the conclusions, ensuring they are congruent with the referenced studies and logically coherent with the paper's arguments.
This paper requires meticulous proofreading to address grammatical or typographical issues. Numerous errors are perceptible within the text, and they can be distracting to the reader. It is recommended that the author seeks the assistance of a professional proofreader to address this concern.
Author Response
We thank the reviewer for the opportunity to improve our paper.
Firstly, this paper requires meticulous proofreading to address grammatical or typographical issues. Numerous errors are perceptible within the text, and they can be distracting to the reader. It is recommended that the author seeks the assistance of a professional proofreader to address this concern.
[Nuno Rodrigues]: Many thanks for the remark, we completely agree. The whole document was proofread. To keep track of the corrections made, we used the track changes feature.
The second noteworthy concern pertains to the structure of the paper, which renders the content and context challenging to grasp, at least from this reviewer's perspective. Despite the extensive use of references, they appear to be used without a clear connection or coherence. As a result, the critical review by the author is not apparent within the paper. To address this issue, the author may consider reorganizing the paper's structure and establishing a clear and logical flow of ideas supported by relevant references.
[Nuno Rodrigues]: Many thanks for the remark, we completely agree. The paper structure was changed (sections were renamed and ideas were reorganized) establishing a clear and logical flow of ideas. To keep track of the corrections made, we used the track changes feature.
The paper's primary research questions were twofold: i) can a clear and definitive path to success be identified when implementing the PPP model in healthcare, particularly in hospitals? ii) is it possible to pinpoint an ideal scenario where the PPP model would unquestionably enhance healthcare quality? However, this reviewer deems that the paper has not fully addressed both inquiries. It is recommended that the author reorient the approach and conduct a comprehensive critical review focusing explicitly on these two questions, enabling them to achieve the research objectives.
[Nuno Rodrigues]: Many thanks for the remark, we completely agree. The manuscript was modified to realign the methodology and allow attention to be given to both research questions. We made advantage of the track modifications tool to record the corrections that were made.
Still related to point c, the first research inquiry is partially answered in the paper's conclusion section, albeit without a clear explanation of its reasoning (i.e., jumping to conclusions). Concerning the second research question, the statements expressed in lines 553-557 are deemed insufficient in their justification and do not adequately respond to the question. The author is advised to revisit the conclusion section and reinforce the arguments with robust reasoning in the discussion section (not in the conclusion section), providing a comprehensive answer to both research questions.
[Nuno Rodrigues]: Many thanks for the remark, we completely agree. The manuscript was modified, including the discussion and conclusion sections. We made advantage of the track modifications tool to record the corrections that were made.
This reviewer further argues that certain conclusions presented in the paper are inconsistent with the cited references and the author's statements. On lines 492-494, the author acknowledged the presence of contradictory findings in the area of hospitals that "leaves open the possibility of there being cases of success and failure, resulting from different political, economic, and social contexts." However, on lines 498-500, the author concluded that "most have shown positive results" regarding healthcare unit performance and cost efficiency. This conclusion appears to be in conflict with the earlier statement. Therefore, it is recommended that the author revisit the conclusions, ensuring they are congruent with the referenced studies and logically coherent with the paper's arguments.
[Nuno Rodrigues]: Many thanks for the remark, we completely agree. The manuscript was modified, including the discussion and conclusion sections. We made advantage of the track modifications tool to record the corrections that were made.
Reviewer 3 Report
This paper conducted a review of the PPP model in hospital industry and tries to find whether it is possible to find the clear path and an ideal scenario when applying the PPP model to healthcare. The data is enough and the topic is interesting. However, there are many questions need to be addressed.
First, the authors do not state clearly their contributions to the literature. This should be added either in the introduction section or in the literature review section. The PPP model in hospital industry is quite normal in literature, what does this paper can add to the previous literature.
Second, the logic of the paper is quite confused. For example, what is the difference between section 4. PPPs and the health sector and section 5. PPPs in health around the world? Why there is a need for section 5. PPPs in health around the world? And what is the meaning of the section 3. Management of the PPP model. I suggest rewriting all the subtitles in the paper and try to figure out what it the main question each section want to answer.
Third, it is very weird that table 1 and Figure 1 are shown in section Conclusions. Generally speaking, we do not give new literature or findings in section Conclusions. I suggest rewriting section Conclusions.
Forth, there are too many grammar errors in the paper. For example, in the abstract, an academic paper usually does not use one. On page 2, “It is possible to identify the clear path to success while applying the PPP model to healthcare (Hospitals)” should be “is it possible to identify the clear path to success while applying the PPP model to healthcare (Hospitals)”. Find a native English speaker to polish the paper.
English very difficult to understand/incomprehensible
Author Response
We thank the reviewers for the opportunity to improve our paper.
First, the authors do not state clearly their contributions to the literature. This should be added either in the introduction section or in the literature review section. The PPP model in hospital industry is quite normal in literature, what does this paper can add to the previous literature.
[Nuno Rodrigues]: Many thanks for the remark, we completely agree. Added to the introduction section. The main contribution is the proposed model that will allow practitioners to evaluate the extent to which the PPP model might be applicable to a specific hospital project.
Second, the logic of the paper is quite confused. For example, what is the difference between section 4. PPPs and the health sector and section 5. PPPs in health around the world? Why there is a need for section 5. PPPs in health around the world? And what is the meaning of the section 3. Management of the PPP model. I suggest rewriting all the subtitles in the paper and try to figure out what it the main question each section want to answer.
[Nuno Rodrigues]: Many thanks for the remark, we completely agree. The paper structure was changed (sections were renamed and ideas were reorganized) establishing a clear and logical flow of ideas. To keep track of the corrections made, we used the track changes feature.
Third, it is very weird that table 1 and Figure 1 are shown in section Conclusions. Generally speaking, we do not give new literature or findings in section Conclusions. I suggest rewriting section Conclusions.
[Nuno Rodrigues]: Many thanks for the remark, we completely agree. The manuscript was modified, including the discussion and conclusion sections. We made advantage of the track modifications tool to record the corrections that were made.
Forth, there are too many grammar errors in the paper. For example, in the abstract, an academic paper usually does not use one. On page 2, “It is possible to identify the clear path to success while applying the PPP model to healthcare (Hospitals)” should be “is it possible to identify the clear path to success while applying the PPP model to healthcare (Hospitals)”. Find a native English speaker to polish the paper.
[Nuno Rodrigues]: Many thanks for the remark, we completely agree. The whole document was proofread. To keep track of the corrections made, we used the track changes feature.
Reviewer 4 Report
The topic “PPP Model Applied to Hospitals” is very meaningful. This is a review paper. However, the writing form can not meet the requirements of the review paper.
(1) The structure of the paper should to be adjusted. The existing structure shows that the logic and hierarchy of the paper are ambiguity.
(2) There are many unclear expressions in the paper. For example, as mentioned in ABSTRACT, “this review proposes a conceptual model for studying the PPP model when applied to Hospitals”. However, this work is only reflected in the conclusion part, and this model is presented without a complete statement. In other words, it doesn't explain why there are 6 dimensions: “(i) Environment; (ii) Potentiate Benefits; (iii) Constant Measure; (iv) Evaluation; (v) Management and (vi) Enhance Strengths.”
(3)Section 3-6, the following related studies are reviewed respectively, “Management of the PPP model, PPPs and the health sector, the performance of the PPP model, PPPs in health around the world”. The authors' work shows only an introduction to existing research. But in the conclusion, the model is presented directly. There is no logical relationship in writing.
(4) The literature review is the writing form of a running book, for example, from line 364 to 486. One paragraph per document, such writing is not appropriate. This also happens in other sections.
The literature review is the writing form of a running book, for example, from line 364 to 486. One paragraph per document, such writing is not appropriate. This also happens in other sections.
Author Response
We thank the reviewers for the opportunity to improve our paper.
(1) The structure of the paper should to be adjusted. The existing structure shows that the logic and hierarchy of the paper are ambiguity.
[Nuno Rodrigues]: Many thanks for the remark, we completely agree. The paper structure was changed (sections were renamed and ideas were reorganized) establishing a clear and logical flow of ideas. To keep track of the corrections made, we used the track changes feature.
(2) There are many unclear expressions in the paper. For example, as mentioned in ABSTRACT, “this review proposes a conceptual model for studying the PPP model when applied to Hospitals”. However, this work is only reflected in the conclusion part, and this model is presented without a complete statement. In other words, it doesn't explain why there are 6 dimensions: “(i) Environment; (ii) Potentiate Benefits; (iii) Constant Measure; (iv) Evaluation; (v) Management and (vi) Enhance Strengths.”
[Nuno Rodrigues]: The manuscript was modified, including discussion and conclusion sections. We made advantage of the track modifications tool to record the corrections that were made.
(3)Section 3-6, the following related studies are reviewed respectively, “Management of the PPP model, PPPs and the health sector, the performance of the PPP model, PPPs in health around the world”. The authors' work shows only an introduction to existing research. But in the conclusion, the model is presented directly. There is no logical relationship in writing.
[Nuno Rodrigues]: Many thanks for the remark, we completely agree. The manuscript was modified, including the discussion and conclusion sections. We made advantage of the track modifications tool to record the corrections that were made.
(4) The literature review is the writing form of a running book, for example, from line 364 to 486. One paragraph per document, such writing is not appropriate. This also happens in other sections.
[Nuno Rodrigues]: Many thanks for the remark, we completely agree. The manuscript was modified, including the writing form. We made advantage of the track modifications tool to record the corrections that were made.
Round 2
Reviewer 2 Report
This reviewer deeply appreciates the author's efforts in addressing the comments provided. Most of the comments have been effectively addressed, encompassing meticulous proofreading and an enhanced writing structure compared to the previous version. Nevertheless, this reviewer contends that the author has not comprehensively responded to their research inquiries. The initial research question concerns discerning a definitive and unambiguous path to success. While the author has indeed proposed a model, they have not expounded upon its inner workings in sufficient detail, as the main aim of this paper is to identify the pathway to success. Similarly, the second question, as previously noted, remains inadequately addressed. It is recommended that the author segregate the discussion and conclusion sections to allow for focused responses to both research questions. Apart from these considerations, no further comments are offered by this reviewer. Should the Editor opt to accept this paper, the suggestions provided herein can be duly considered to enhance its overall quality.
Overall, the text is commendable, although a few minor errors can still be detected.
Author Response
We thank the reviewer for the opportunity to improve our paper. Sections Discussion and conclusion are defined together since most critical reviews are set in this way, mostly because the discussion is based on the existing reviews combined with a conclusion drawn from the knowledge already established. The model is explained throughout this section being stated how the path to success might be connected to the ideal scenario for PPP projects implementation in hospitals. Minor corrections were conducted in order to address the minor errors detected.
Once again we would like to value the reviewer's strong contribution and comments.
Reviewer 3 Report
the author has revised the manuscript extensively based on the comments.
need improvement
Author Response
We thank the reviewer for the opportunity to improve our paper.
Minor corrections were conducted in order to address the English errors detected.
Once again we would like to value the reviewer's strong contribution and comments.
Reviewer 4 Report
(1)Section 3-6, the related studies are reviewed respectively, but he authors' work shows only an introduction to existing research. For each section, there is no relevant literature review.
(2) In the literature review, some parts are introduced one by one. The author's point of view is missing.
none
Author Response
We thank the reviewer for the opportunity to improve our paper.
The literature review was elaborated based and approached taking into consideration a state-of-the-art scope. The author's point of view was established in section 6. Minor changes were conducted to address the suggestions.
Once again we would like to value the reviewer's strong contribution and comments.